# Graph Clustering: Block-models and model free results

**Yali Wan**
Department of Statistics
University of Washington
Seattle, WA 98195-4322, USA
`yaliwan@washington.edu`

**Marina Meilă**
Department of Statistics
University of Washington
Seattle, WA 98195-4322, USA
`mmp@stat.washington.edu`

## Abstract

Clustering graphs under the Stochastic Block Model (SBM) and extensions are well studied. Guarantees of correctness exist under the assumption that the data is sampled from a model. In this paper, we propose a framework, in which we obtain "correctness" guarantees *without assuming the data comes from a model*. The guarantees we obtain depend instead on the statistics of the data that can be checked. We also show that this framework ties in with the existing model-based framework, and that we can exploit results in model-based recovery, as well as strengthen the results existing in that area of research.

## 1 Introduction: a framework for clustering with guarantees without model assumptions

In the last few years, model-based clustering in networks has witnessed spectacular progress. At the central of intact are the so-called *block-models*, the *Stochastic Block Model (SBM)*, *Degree-Corrected SBM (DC-SBM)* and *Preference Frame Model (PFM)*. The understanding of these models has been advanced, especially in understanding the conditions when recovery of the true clustering is possible with small or no error. The algorithms for recovery with guarantees have also been improved. However, the impact of the above results is limited by the assumption that the observed data comes from the model.

This paper proposes a framework to provide *theoretical guarantees for the results of model based clustering algorithms, without making any assumption about the data generating process*. To describe the idea, we need some notation. Assume that a graph $\mathcal{G}$ on $n$ nodes is observed. A model-based algorithm clusters $\mathcal{G}$, and outputs clustering $\mathcal{C}$ and parameters $\mathcal{M}(\mathcal{G}, \mathcal{C})$.

The framework is as follows: if $\mathcal{M}(\mathcal{G}, \mathcal{C})$ fits the data $\mathcal{G}$ well, then we shall prove that any other clustering $\mathcal{C}'$ of $\mathcal{G}$ that also fits $\mathcal{G}$ well will be a small perturbation of $\mathcal{C}$. If this holds, then $\mathcal{C}$ with model parameters $\mathcal{M}(\mathcal{G}, \mathcal{C})$ can be said to capture the data structure in a meaningful way.

We exemplify our approach by obtaining model-free guarantees for the SBM and PFM models. Moreover, we show that model-free and model-based results are intimately connected.

## 2 Background: graphs, clusterings and block models

**Graphs, degrees, Laplacian, and clustering**   Let $\mathcal{G}$ be a graph on $n$ nodes, described by its *adjacency matrix* $\hat{A}$. Define $\hat{d}_i = \sum_{j=1}^{n} \hat{A}_{ij}$ the *degree* of node $i$, and $\hat{D} = \mathrm{diag}\{\hat{d}_i\}$ the diagonal matrix of the node degrees. The *(normalized) Laplacian* of $\mathcal{G}$ is defined as[1] $\hat{L} = \hat{D}^{-1/2} \hat{A} \hat{D}^{-1/2}$. In

extension, we define the *degree matrix* $D$ and the Laplacian $L$ associated to any matrix $A \in \mathbb{R}^{n \times n}$, with $A_{ij} = A_{ji} \geq 0$, in a similar way.

Let $\mathcal{C}$ be a partitioning (clustering) of the nodes of $\mathcal{G}$ into $K$ clusters. We use the shorthand notation $i \in k$ for "node $i$ belongs to cluster $k$". We will represent $\mathcal{C}$ by its $n \times K$ *indicator matrix* $Z$, defined by

$$Z_{ik} = 1 \text{ if } i \in k, 0 \text{ otherwise, for } i = 1, \ldots n, \; k = 1, \ldots K. \tag{1}$$

Note that $Z^T Z = \mathrm{diag}\{n_k\}$ with $n_k$ counting the number of nodes in cluster $k$, and $Z^T \hat{A} Z = [n_{kl}]_{k,l=1}^K$ with $n_{kl}$ counting the edges in $\mathcal{G}$ between clusters $k$ and $l$. Moreover, for two indicator matrices $Z, Z'$ for clusterings $\mathcal{C}, \mathcal{C}'$, $(Z^T Z')_{kk'}$ counts the number of points in the intersection of cluster $k$ of $\mathcal{C}$ with cluster $k'$ of $\mathcal{C}'$, and $(Z^T \hat{D} Z')_{kk'}$ computes $\sum_{i \in k \cap k'} \hat{d}_i$ the volume of the same intersection.

**"Block models" for random graphs (SBM, DC-SBM, PFM)** This family of models contains Stochastic Block Models (SBM) [1, 18], Degree-Corrected SBM (DC-SBM) [17] and Preference Frame Models (PFM) [20]. Under each of these model families, a graph $\mathcal{G}$ with adjacency matrix $\hat{A}$ over $n$ nodes is generated by sampling its edges *independently* following the law $\hat{A}_{ij} \sim Bernoulli(A_{ij})$, for all $i > j$. The symmetric matrix $A = [A_{ij}]$ describing the graph is the *edge probability matrix*. The three model families differ in the constraints they put on an acceptable $A$. Let $\mathcal{C}^*$ be a clustering. The entries of $A$ are defined w.r.t $\mathcal{C}^*$ as follows (and we say that $A$ is *compatible* with $\mathcal{C}^*$).

SBM : $A_{ij} = B_{kl}$ whenever $i \in k, j \in l$, with $B = [B_{kl}] \in \mathbb{R}^{K \times K}$ symmetric and non-negative.

DC-SBM : $A_{ij} = w_i w_j B_{kl}$ whenever $i \in k, j \in l$, with $B$ as above and $w_1, \ldots w_n$ non-negative weights associated with the graph nodes.

PFM : $A$ satisfies $D = \mathrm{diag}(A\mathbf{1})$, $D^{-1}AZ = ZR$ where $\mathbf{1}$ denotes the vector of all ones, $Z$ is the indicator matrix of $\mathcal{C}^*$, and $R$ is a stochastic matrix ($R\mathbf{1} = \mathbf{1}$, $R_{kl} \geq 0$), the details are in [20]

While perhaps not immediately obvious, the SBM is a subclass of the DC-SBM, and the latter a subclass of the PFM. Another common feature of block-models, that will be significant throughout this work is that for all three, Spectral Clustering algorithms [15] have been proved to work well estimating $\mathcal{C}^*$.

## 3 Main theorem: blueprint and results for PFM, SBM

Let $\mathcal{M}$ be a model class, such as SBM, DC-SBM, PFM, and denote $\mathcal{M}(\mathcal{G}, \mathcal{C}) \in \mathcal{M}$ to be a model that is compatible with $\mathcal{C}$ and is fitted in some way to graph $\mathcal{G}$ (we do not assume in general that this fit is optimal).

**Theorem 1 (Generic Theorem)** *We say that clustering $\mathcal{C}$ fits $\mathcal{G}$ well w.r.t $\mathcal{M}$ iff $\mathcal{M}(\mathcal{G}, \mathcal{C})$ is "close to" $\mathcal{G}$. If $\mathcal{C}$ fits $\mathcal{G}$ well w.r.t $\mathcal{M}$, then (subject to other technical conditions) any other clustering $\mathcal{C}'$ which also fits $\mathcal{G}$ well is close to $\mathcal{C}$, i.e. $\mathrm{dist}(\mathcal{C}, \mathcal{C}')$ is small.*

In what follows, we will instantiate this Generic Theorem, and the concepts therein; in particular the following will be formally defined. (1) Model construction, i.e an algorithm to fit a model in $\mathcal{M}$ to $(\mathcal{G}, \mathcal{C})$. This is necessary since we want our results to be computable in practice. (2) A goodness of fit measure between $\mathcal{M}(\mathcal{C}, \mathcal{G})$ and the data $\mathcal{G}$. (3) A distance between clusterings. We adopt the widely used *Misclassification Error (or Hamming)* distance defined below.

The *Misclassification Error (ME) distance* between two clusterings $\mathcal{C}, \mathcal{C}'$ over the same set of $n$ points is

$$\mathrm{dist}(\mathcal{C}, \mathcal{C}') = 1 - \frac{1}{n} \max_{\pi \in \mathbb{S}_K} \sum_{i \in k \cap \pi(k)} 1, \tag{2}$$

where $\pi$ ranges over all permutations of $K$ elements $\mathbb{S}_K$, and $\pi(k)$ indexes a cluster in $\mathcal{C}'$. If the points are weighted by their degrees, a natural measure on the node set, the *Weighted ME (wME)*

*distance* is

$$\text{dist}_{\hat{d}}(\mathcal{C}, \mathcal{C}') = 1 - \frac{1}{\sum_{i=1}^{n} \hat{d}_i} \max_{\pi \in \mathbb{S}_K} \sum_{i \in k \cap \pi(k)} \hat{d}_i. \tag{3}$$

In the above, $\sum_{i \in k \cap k'} \hat{d}_i$ represents the total weight of the set of points assigned to cluster $k$ by $\mathcal{C}$ and to cluster $k'$ ( or $\pi(k)$) by $\mathcal{C}'$. Note that in the indicator matrix representation of clusterings, this is the $(k, k')$ element of the matrix $Z^T \hat{D} Z' \in \mathbb{R}^{K \times K}$. While dist is more popular, we believe $\text{dist}_{\hat{d}}$ is more natural, especially when node degrees are dissimilar, as $\hat{d}$ can be seen as a natural measure on the set of nodes, and $\text{dist}_{\hat{d}}$ is equivalent to the *earth-mover's* distance.

## 3.1 Main result for PFM

**Constructing a model** Given a graph $\mathcal{G}$ and a clustering $\mathcal{C}$ of its nodes, we wish to construct a PFM compatible with $\mathcal{C}$, so that its Laplacian $L$ satisfies that $||\hat{L} - L||$ is small.

Let the spectral decomposition of $\hat{L}$ be

$$\hat{L} = [\hat{Y} \ \hat{Y}_{low}] \begin{bmatrix} \hat{\Lambda} & 0 \\ 0 & \hat{\Lambda}_{low} \end{bmatrix} \begin{bmatrix} \hat{Y}^T \\ \hat{Y}_{low}^T \end{bmatrix} = \hat{Y} \hat{\Lambda} \hat{Y}^T + \hat{Y}_{low} \hat{\Lambda}_{low} \hat{Y}_{low}^T \tag{4}$$

where $\hat{Y} \in \mathbb{R}^{n \times K}, \hat{Y}_{low} \in \mathbb{R}^{n \times (n-K)}, \hat{\Lambda} = diag(\hat{\lambda}_1, \cdots, \hat{\lambda}_K), \hat{\Lambda}_{low} = diag(\hat{\lambda}_{K+1}, \cdots, \hat{\lambda}_n)$. To ensure that the matrices $\hat{Y}, \hat{Y}_{low}$ are uniquely defined we assume throughout the paper that $\hat{L}$'s $K$-th eigengap, i.e, $|\lambda_K| - |\lambda_{K+1}|$, is non-zero.

**Assumption 1** *The eigenvalues of $\hat{L}$ satisfy $\hat{\lambda}_1 = 1 \geq |\hat{\lambda}_2| \geq \ldots \geq |\hat{\lambda}_K| > |\hat{\lambda}_{K+1}| \geq \ldots |\hat{\lambda}_n|$.*

Denote the subspace spanned by the columns of $M$, for any $M$ matrix, by $\mathcal{R}(M)$, and $|| \ ||$ the Euclidean or spectral norm.

---

PFM Estimation Algorithm

**Input** Graph $\mathcal{G}$ with $\hat{A}, \hat{D}, \hat{L}, \hat{Y}, \hat{\Lambda}$, clustering $\mathcal{C}$ with indicator matrix $Z$.

**Output** $(A, L) = PFM(\mathcal{G}, \mathcal{C})$

1. Construct an orthogonal matrix derived from $Z$.

$$Y_Z = \hat{D}^{1/2} Z C^{-1/2}, \text{ with } C = Z^T \hat{D} Z \text{ the column normalization of } Z. \tag{5}$$

Note $C_{kk} = \sum_{i \in k} \hat{d}_i$ is the volume of cluster $k$.

2. Project $Y_Z$ on $\hat{Y}$ and perform Singular Value Decomposition.

$$F = Y_Z^T \hat{Y} = U \Sigma V^T \tag{6}$$

3. Change basis in $\mathcal{R}(Y_Z)$ to align with $\hat{Y}$.

$$Y = Y_Z U V^T. \quad \text{Complete } Y \text{ to an orthonormal basis } [Y \ B] \text{ of } \mathbb{R}^n. \tag{7}$$

4. Construct Laplacian $L$ and edge probability matrix $A$.

$$L = Y \hat{\Lambda} Y^T + (BB^T) \hat{L} (BB^T), \qquad A = \hat{D}^{1/2} L \hat{D}^{1/2}. \tag{8}$$

---

**Proposition 2** *Let $\mathcal{G}, \hat{A}, \hat{D}, \hat{L}, \hat{Y}, \hat{\Lambda}$ and $Z$ be defined as above, and $(A, L) = PFM(\mathcal{G}, \mathcal{C})$. Then,*

1. *$\hat{D}$ and L, or A define a PFM with degrees $\hat{d}_{1:n}$.*

2. *The columns of Y are eigenvectors of L with eigenvalues $\hat{\lambda}_{1:K}$.*

3. *$\hat{D}^{1/2} \mathbf{1}$ is an eigenvector of both L and $\hat{L}$ with eigenvalue $\hat{\lambda}_1 = 1$.*

The proof is relegated to the Supplement, as are all the omitted proofs.

$PFM(\mathcal{G}, \mathcal{C})$ is an estimator for the PFM parameters given the clustering. It is evidently not the Maximum Likelihood estimator, but we can show that it is consistent in the following sense.

**Proposition 3 (Informal)** *Assume that $\mathcal{G}$ is sampled from a PFM with parameters $D^*, L^*$ and compatible with $\mathcal{C}^*$, and let $L = PFM(\mathcal{G}, \mathcal{C}^*)$. Then, under standard recovery conditions for PFM (e.g [20]) $||L^* - L|| = o(1)$ w.r.t. $n$.*

**Assumption 2 (Goodness of fit for PFM)** $||\hat{L} - L|| \leq \varepsilon$.

$PFM(\mathcal{G}, \mathcal{C})$ instantiates $\mathcal{M}(\mathcal{G}, \mathcal{C})$, and Assumption 2 instantiates the goodness of fit measure. It remains to prove an instance of Generic Theorem 1 for these choices.

**Theorem 4 (Main Result (PFM))** *Let $\mathcal{G}$ be a graph with $\hat{d}_{1:n}, \hat{D}, \hat{L}, \hat{\lambda}_{1:n}$ as defined, and $\hat{L}$ satisfy Assumption 1. Let $\mathcal{C}, \mathcal{C}'$ be two clusterings with $K$ clusters, and $L, L'$ be their corresponding Laplacians, defined as in (8), and satisfy Assumption 2 respectively. Set $\delta = \frac{(K-1)\varepsilon^2}{(|\hat{\lambda}_K| - |\hat{\lambda}_{K+1}|)^2}$ and $\delta_0 = \min_k C_{kk} / \max_k C_{kk}$ with $C$ defined as in (5), where $k$ indexes the clusters of $\mathcal{C}$. Then, whenever $\delta \leq \delta_0$,*

$$\operatorname{dist}_{\hat{d}}(\mathcal{C}, \mathcal{C}') \leq \frac{\max_k C_{kk}}{\sum_k C_{kk}} \delta, \tag{9}$$

*with $\operatorname{dist}_{\hat{d}}$ being the weighted ME distance (3).*

In the remainder of this section we outline the proof steps, while the partial results of Proposition 5, 6, 7 are proved in the Supplement. First, we apply the perturbation bound called the Sinus Theorem of Davis and Kahan, in the form presented in Chapter V of [19].

**Proposition 5** *Let $\hat{Y}, \hat{\lambda}_{1:n}, Y$ be defined as usual. If Assumptions 1 and 2 hold, then*

$$|| \operatorname{diag}(\sin \theta_{1:K}(\hat{Y}, Y))|| \leq \frac{\varepsilon}{|\hat{\lambda}_K| - |\hat{\lambda}_{K+1}|} = \varepsilon' \tag{10}$$

*where $\theta_{1:K}$ are the canonical (or principal) angles between $\mathcal{R}(\hat{Y})$ and $\mathcal{R}(Y)$ (see e.g [8]).*

The next step concerns the closeness of $Y, \hat{Y}$ in Frobenius norm. Since Proposition 5 bounds the sinuses of the canonical angles, we exploit the fact that the cosines of the same angles are the singular values of $F = Y^T \hat{Y}$ of (6).

**Proposition 6** *Let $M = YY^T$, $\hat{M} = \hat{Y}\hat{Y}^T$ and $F, \varepsilon'$ as above. Assumptions 1 and 2 imply that*

1. *$||F||_F^2 = \operatorname{trace} M\hat{M}^T \geq K - (K-1)\varepsilon'^2$.*

2. *$||M - \hat{M}||_F^2 \leq 2(K-1)\varepsilon'^2$.*

Now we show that all clusterings which satisfy Proposition 6 must be close to each other in the weighted ME distance. For this, we first need an intermediate result. Assume we have two clusterings $\mathcal{C}, \mathcal{C}'$, with $K$ clusters, for which we construct $Y_Z, Y, L, M$, respectively $Y_Z', Y', L', M'$ as above. Then, the subspaces spanned by $Y$ and $Y'$ will be close.

**Proposition 7** *Let $\hat{L}$ satisfy Assumption 1 and let $\mathcal{C}, \mathcal{C}'$ represent two clusterings for which $L, L'$ satisfy Assumption 2. Then, $||Y_Z^T Y_Z'||_F^2 \geq K - 4(K-1)\varepsilon'^2 = K - \delta$*

The main result now follows from Proposition 7 and Theorem 9 of [13], as shown in the Supplement. This proof approach is different from the existing perturbation bounds for clustering, which all use counting arguments. The result of [13] is a *local* equivalence, which bounds the error we need in terms of $\delta$ defined above ("local" meaning the result only holds for small $\delta$).

## 3.2 Main Theorem for SBM

In this section, we offer an instantiation of Generic Theorem 1 for the case of the SBM. As before, we start with a model estimator, which in this case is the Maximum Likelihood estimator.

---
SBM Estimation Algorithm

**Input** Graph with $\hat{A}$, clustering $\mathcal{C}$ with indicator matrix $Z$.

**Output** $A = SBM(\mathcal{G}, \mathcal{C})$

1. Construct an orthogonal matrix derived from $Z$: $Y_Z = ZC^{-1/2}$ with $C = Z^T Z$.

2. Estimate the edge probabilities: $B = C^{-1} Z^T \hat{A} Z C^{-1}$.

3. Construct $A$ from $B$ by $A = ZBZ^T$.

---

**Proposition 8** *Let $\tilde{B} = C^{1/2} B C^{1/2}$ and denote the eigenvalues of $\tilde{B}$, ordered by decreasing magnitude, by $\lambda_{1:K}$. Let the spectral decomposition of $\tilde{B}$ be $\tilde{B} = U \Lambda U^T$, with $U$ an orthogonal matrix and $\Lambda = \mathrm{diag}(\lambda_{1:K})$. Then*

1. *$A$ is a SBM.*

2. *$\lambda_{1:K}$ are the $K$ principal eigenvalues of $A$. The remaining eigenvalues of $A$ are zero.*

3. *$A = Y \Lambda Y^T$ where $Y = Y_Z U$.*

**Assumption 3 (Eigengap)** *$B$ is non-singular (or, equivalently, $|\lambda_K| > 0$.*

**Assumption 4 (Goodness of fit for SBM)** *$||\hat{A} - A|| \leq \varepsilon$.*

With the model (SBM), estimator, and goodness of fit defined, we are ready for the main result.

**Theorem 9 (Main Result (SBM))** *Let $\mathcal{G}$ be a graph with incidence matrix $\hat{A}$, and $\hat{\lambda}_K^A$ the $K$-th singular value of $\hat{A}$. Let $\mathcal{C}, \mathcal{C}'$ be two clusterings with $K$ clusters, satisfying Assumptions 3 and 4. Set $\delta = \frac{4K\varepsilon^2}{|\hat{\lambda}_K^A|^2}$ and $\delta_0 = \min_k n_k / \max_k n_k$, where $k$ indexes the clusters of $\mathcal{C}$. Then, whenever $\delta \leq \delta_0$, $\mathrm{dist}(\mathcal{C}, \mathcal{C}') \leq \delta \max_k n_k / n$, where* dist *represents the ME distance* (2).

Note that the eigengap of $\hat{A}$, $\hat{\Lambda}_K^A$ is not bounded above, and neither is $\varepsilon$. Since the SBM is less flexible than the PFM, we expect that for the same data $\mathcal{G}$, Theorem 9 will be more restrictive than Theorem 4.

# 4 The results in perspective

## 4.1 Cluster validation

Theorems like 4, 9 can provide model free guarantees for clustering. We exemplify this procedure in the experimental Section 6, using standard spectral clustering as described in e.g [18, 17, 15]. What is essential is that all the quantities such as $\varepsilon$ and $\delta$ *are computable from the data*.

Moreover, if $Y$ is available, then the bound in Theorem 4 can be improved.

**Proposition 10** *Theorem 4 holds when $\delta$ is replaced by $\delta_Y = K - <\hat{M}, M>_F + (K-1)(\varepsilon')^2 + 2\sqrt{2(K-1)}\varepsilon'||\hat{M} - M||_F$, with $\varepsilon' = \varepsilon/(|\hat{\lambda}_K| - |\hat{\lambda}_{K+1}|)$ and $M, \hat{M}$ defined in Proposition 6.*

## 4.2 Using existing model-based recovery theorems to prove model-free guarantees

We exemplify this by using (the proof of) Theorem 3 of [20] to prove the following.

**Theorem 11 (Alternative result based on [20] for PFM)** *Under the same conditions as in Theorem 4,* $\mathrm{dist}_{\hat{d}}(\mathcal{C}, \mathcal{C}') \leq \delta_{WM}$, *with $\delta_{WM} = 128 \frac{K\varepsilon^2}{(|\hat{\lambda}_K| - |\hat{\lambda}_{K+1}|)^2}$.*

It follows, too, that with the techniques in this paper, the error bound in [20] can be improved by a factor of 128.

Similarly, if we use the results of [18] we obtain alternative model-free guarantee for the SBM.

**Assumption 5 (Alternative goodness of fit for SBM)** $||\hat{L}^2 - L^2||_F \leq \varepsilon$, *where $\hat{L}$, $L$ are the Laplacians of $\hat{A}$ and $A = SBM(\mathcal{G}, \mathcal{C})$ respectively.*

**Theorem 12 (Alternative result based on [18] for SBM)** *Under the same conditions as in Theorem 9, except for replacing Assumption 4 with 5, $\text{dist}(\mathcal{C}, \mathcal{C}') \leq \delta_{RCY}$ with $\delta_{RCY} = \frac{\varepsilon^2}{|\hat{\lambda}_K|^4} \frac{16 \max_k n_k}{n}$.*

A problem with this result is that Assumption 5 is much stronger than 4 (being in Frobenius norm). The more recent results of [17] (with unspecified constants) in conjunction with our original Assumptions 3, 4, and the assumption that all clusters have equal sizes, give a bound of $\mathcal{O}(K\varepsilon^2/\hat{\lambda}_K^2)$ for the SBM; hence our model-free Theorem 9 matches this more restrictive model-based theorem.

### 4.3 Sanity checks and Extensions

It can be easily verified that if indeed $\mathcal{G}$ is sampled from a SBM, or PFM, then for large enough $n$, and large enough model eigengap, Assumptions 1 and 2 (or 3 and 4) will hold.

Some immediate extensions and variations of Theorems 4, 9 are possible. For example, one could replace the spectral norm by the Frobenius norm in Assumptions 2 and 4, which would simplify some of the proofs. However, using the Frobenius norm would be a much stronger assumption [18] Theorem 4 holds not just for simple graphs, but in the more general case when $\hat{A}$ is a weighted graph (i.e. a *similarity matrix*). The theorems can be extended to cover the case when $\mathcal{C}'$ is a clustering that is $\alpha$-worse than $\mathcal{C}$, i.e when $||L' - \hat{L}|| \geq ||L - \hat{L}||(1 - \alpha)$.

### 4.4 Clusterability and resilience

Our Theorems also imply the stability of a clustering to perturbations of the graph $\mathcal{G}$. Indeed, let $\hat{L}'$ be the Laplacian of $\mathcal{G}'$, a perturbation of $\mathcal{G}$. If $||\hat{L}' - \hat{L}|| \leq \varepsilon$, then $||\hat{L}' - L|| \leq 2\varepsilon$, and (1) $\mathcal{G}'$ is well fitted by a PFM whenever $\mathcal{G}$ is, and (2) $\mathcal{C}$ is $\delta$ stable w.r.t $\mathcal{G}'$, hence $\mathcal{C}$ is what some authors [9] call *resilient*.

A graph $\mathcal{G}$ is *clusterable* when $\mathcal{G}$ can be fitted well by some clustering $\mathcal{C}^*$. Much work [4, 7] has been devoted to showing that clusterability implies that finding a $\mathcal{C}$ close to $\mathcal{C}^*$ is computationally efficient. Such results can be obtained in our framework, by exploiting existing recovery theorems such as [18, 17, 20], which give recovery guarantees for Spectral Clustering, under the assumption of sampling from the model. For this, we can simply replace the model assumption with the assumption that there is a $\mathcal{C}^*$ for which $L$ (or $A$) satisfies Assumptions 1 and 2 (or 3 and 4).

## 5 Related work

To our knowledge, there is no work of the type of Theorem 1 in the literature on SBM, DC-SBM, PFM. The closest work is by [6] which guarantees approximate recovery *assuming $\mathcal{G}$ is close to a DC-SBM.*

Spectral clustering is also used for loss-based clustering in (weighted) graphs and some stability results exist in this context. Even though they measure clustering quality by different criteria, so that the $\varepsilon$ values are not comparable, we review them here. The recent paper of [16], Theorem 1.2 states that if the $K$-way *Cheeger constant* of $\mathcal{G}$ is $\rho(k) \leq (1 - \hat{\lambda}_{K+1})/(cK^3)$ then the clustering error[2] $\text{dist}_{\hat{d}}(\mathcal{C}, \mathcal{C}^{opt}) \leq C/c = \delta_{PSZ}$. In the current proof, the constant $C = 2 \times 10^5$; moreover, $\rho(K)$ cannot be computed tractably. In [14], the bound $\delta_{MSX}$ depends on $\varepsilon_{MSX}$, the *Normalized Cut* scaled by the eigengap. Since both bounds refer to the result of spectral clustering, we can compare the relationship between $\delta_{MSX}$ and $\varepsilon_{MSX}$; for [14], this is $\delta_{MSX} = 2\varepsilon_{MSX}[1 - \varepsilon_{MSX}/(K - 1)]$,

which is about $K-1$ times larger than $\delta$ when $\epsilon = \epsilon_{MSX}$. In [5], $\mathrm{dist}(\mathcal{C},\mathcal{C}')$ is defined in terms of $||Y_Z^T - Y_Z'||_F^2$, and the loss is (closely related) to $||\hat{A} - SBM(\mathcal{G},\mathcal{C})||_F^2$. The bound does not take into account the eigengap, that is, the stability of the subspace $\hat{Y}$ itself.

Bootstrap for validating a clustering $\mathcal{C}$ was studied in [11] (see also references therein for earlier work). In [3] the idea is to introduce a statistics, and large deviation bounds for it, *conditioned on sampling from a SBM* (with covariates) and on a given $\mathcal{C}$.

## 6   Experimental evaluation

**Experiment Setup** Given $\mathcal{G}$, we obtain a clustering $\mathcal{C}_0$ by spectral clustering [15]. Then we calculate clustering $\mathcal{C}$ by perturbing $\mathcal{C}_0$ with gradually increasing noise. For each $\mathcal{C}$, we construct PFM $(\mathcal{C},\mathcal{G})$and SBM$(\mathcal{C},\mathcal{G})$ model, and further compute $\epsilon$, $\delta$ and $\delta_0$. If $\delta \le \delta_0$, $\mathcal{C}$ is guaranteed to be stable by the theorems. In the remainder of this section, we describe the data generating process for the simulated datasets and the results we obtained.

**PFM Datasets** We generate from PFM model with $K = 5$, $n = 10000$, $\lambda_{1:K} = (1, 0.875, 0.75, 0.625, 0.5)$. $eigengap = 0.48$, $n_{1:K} = (2000, 2000, 2000, 2000, 2000)$. The stochastic matrix $R$ and its stationary distribution $\rho$ are shown below. We sample an adjacency matrix $\hat{A}$ from $A$ (shown below).

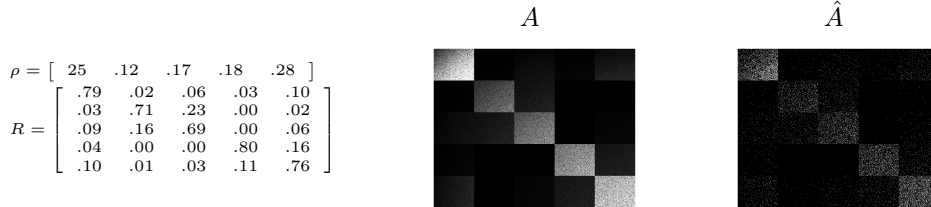

$$\rho = \begin{bmatrix} 25 & .12 & .17 & .18 & .28 \end{bmatrix}$$

$$R = \begin{bmatrix} .79 & .02 & .06 & .03 & .10 \\ .03 & .71 & .23 & .00 & .02 \\ .09 & .16 & .69 & .00 & .06 \\ .04 & .00 & .00 & .80 & .16 \\ .10 & .01 & .03 & .11 & .76 \end{bmatrix}$$

**Perturbed PFM Datasets** $A$ is obtained from the previous model by perturbing its principal subspace (details in Supplement). Then we sample $\hat{A}$ from $A$.

**Lancichinetti-Fortunato-Radicchi (LFR) simulated matrix [12]** The LFR benchmark graphs are widely used for community detection algorithms, due to heterogeneity in the distribution of node degree and community size. A LFR matrix is simulated with $n = 10000$, $K = 4$, $n_k = (2467, 2416, 2427, 2690)$ and $\mu = 0.2$, where $\mu$ is the mixing parameter indicating the fraction of edges shared between a node and the other nodes from outside its community.

**Political Blogs Dataset** A directed network $\vec{A}$ of hyperlinks between weblogs on US politics, compiled from online directories by Adamic and Glance [2], where each blog is assigned a political leaning, liberal or conservative, based on its blog content. The network $A$ contains 1490 blogs. After erasing the disconnected nodes, $n = 983$. We study $\hat{A} = (\vec{A}^T \vec{A})^3$, which is a smoothed undirected graph. For $\vec{A}^T \vec{A}$ we find no guarantees.

The first two data sets are expected to fit the PFM well, but not the SBM, while the LFR data is expected to be a good fit for a SBM. Since all bounds can be computed on weighted graphs as well, we have run the experiments also on the edge probability matrices $A$ used to generate the PFM and perturbed PFM graphs.

The results of these experiments are summarized in Figure 1. For all of the experiments, the clustering $\mathcal{C}$ is ensured to be stable by Theorem 4 as the unweighted error grows to a breaking point, then the assumptions of the theorem fail. In particular, the $\mathcal{C}_0$ is always stable in the PFM framework.

Comparing $\delta$ from Theorem 9 to that from Theorem 4, we find that Theorem 9 (guarantees for SBM) is much harder to satisfy. All $\delta$ values from Theorem 9 are above 1, and not shown.[3] In particular, for the SBM model class, the $\mathcal{C}$ cannot be proved stable even for the LFR data.

Note that part of the reason why with the PFM model very little difference from the clustering $\mathcal{C}_0$ can be tolerated for a clustering to be stable is that the large eigengap makes $PFM(\mathcal{G}, \mathcal{C})$ differ from $PFM(\mathcal{G}, \mathcal{C}_0)$ even for very small perturbations. By comparing the bounds for $\hat{A}$ with the bounds for the "weighted graphs" $A$, we can evaluate that the sampling noise on $\delta$ is approximately equal to that of the clustering perturbation. Of course, the sampling noise varies with $n$, decreasing for larger graphs. Moreover, from Political Blogs data, we see that "smoothing" a graph, by e.g. taking powers of its adjacency matrix, has a stability inducing effect.

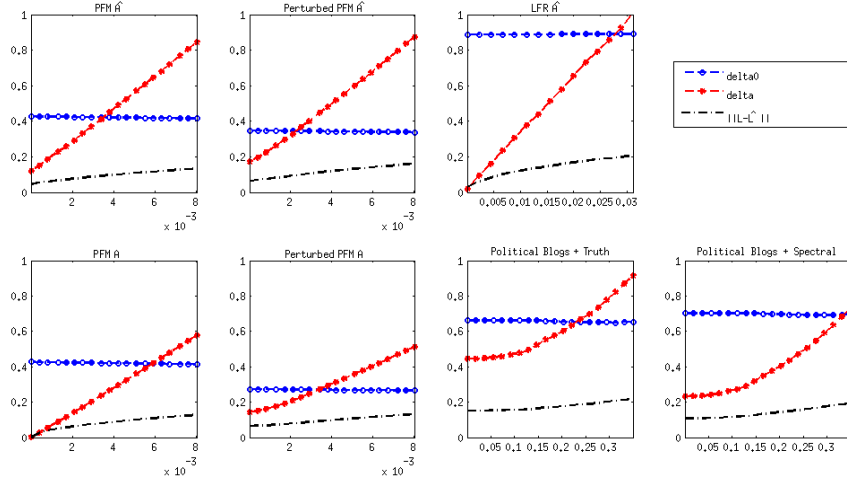

**Figure 1:** Quantities $\epsilon$, $\delta$, $\delta_0$ from Theorem 4 plotted vs $\mathrm{dist}(\mathcal{C}, \mathcal{C}_0)$ for various datasets: $\hat{A}$ denotes a simple graph, while $A$ denotes a weighted graph (i.e. a non-negative matrix). For the Political Blogs: Truth means $\mathcal{C}_0$ is true clustering of [2], spectral means $\mathcal{C}_0$ is obtained from spectral clustering. For SBM, $\delta$ is always greater than $\delta_0$.

## 7  Discussion

This paper makes several contributions. At a high level, it poses the problem of model free validation in the area of community detection in networks. The stability paradigm is not entirely new, but using it explicitly with model-based clustering (instead of cost-based) is. So is "turning around" the model-based recovery theorems to be used in a model-free framework.

All quantities in our theorems are computable from the data and the clustering $\mathcal{C}$, i.e do not contain undetermined constants, and do not depend on parameters that are not available. As with distribution-free results in general, making fewer assumptions allows for less confidence in the conclusions, and the results are not always informative. Sometimes this should be so, e.g when the data does not fit the model well. But it is also possible that the fit is good, but not good enough to satisfy the conditions of the theorems as they are currently formulated. This happens with the SBM bounds, and we believe tighter bounds are possible for this model. It would be particularly interesting to study the non-spectral, sharp thresholds of [1] from the point of view of model-free recovery. A complementary problem is to obtain *negative guarantees* (i.e that $\mathcal{C}$ is *not* unique up to perturbations).

At the technical level, we obtain several different and model-specific stability results, that bound the perturbation of a clustering by the perturbation of a model. They can be used both in model-free and in existing or future model-based recovery guarantees, as we have shown in Section 3 and in the experiments. The proof techniques that lead to these results are actually simpler, more direct, and more elementary than the ones found in previous papers.

## Footnotes

[1]Rigorously speaking, the normalized graph Laplacian is $I - \hat{L}$ [10].

[2]The results is stronger, bounding the perturbation of each cluster individually by $\delta_{PSZ}$, but it also includes a factor larger than 1, bounding the error of $K$-means algorithm.

[3]We also computed $\delta_{RCY}$ but the bounds were not informative.

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
