[Supplementary Material]

# 8 Supplementary Material for Graph Clustering: Block-models and model free results

**Proof of Proposition 2**

1. Proof by verification.
2. $LY = Y\hat{\Lambda}Y^T Y + (BB^T)\hat{L}(BB^T)Y = Y\hat{\Lambda}$. Since $B$ is the orthogonal complement of $Y$, it follows that it is a stable subspace as well.
3. This is a well known result; see for example [19].

The celebrated Sinus Theorem is reproduced here for completeness.

**Theorem 13 (Sinus Theorem of Davis-Kahan, from [19], Theorem V.3.6)** *Let $\hat{L}$ be a Hermitian matrix with spectral resolution given by (4), $Y$ be any $n \times K$ matrix with orthonormal columns, and $M$ any symmetric $K \times K$ matrix with eigenvalues $\mu_{1:K}$. Let $R = \hat{L}Y - YM$ and $\Delta = \min_{\lambda \in \hat{\lambda}_{K+1:n}, \mu \in \mu_{1:K}} |\lambda - \mu| > 0$. Then, for any unitarily invariant norm $||\ ||$, $||\operatorname{diag}(\sin\theta_{1:K}(\hat{Y}, Y))|| \leq \frac{||R||}{\Delta}$, where $\theta_{1:K}$ are the canonical angles between $\mathcal{R}(\hat{Y})$ and $\mathcal{R}(Y)$.*

**Proof of Proposition 5** This is a corollary of Theorem 3.6 in [19]. If eigenvalues are sorted by their absolute values, then $\hat{\lambda}_{K+1:n} \in [-|\hat{\lambda}_{K+1}|, |\hat{\lambda}_{K+1}|]$ and $\mu_{1:K} \in \mathbb{R} \setminus (-|\hat{\lambda}_{K+1}| - \Delta, |\hat{\lambda}_{K+1}| + \Delta)$. If we set $M = \hat{\Lambda}$, so that $\hat{\lambda}_{1:K} \in \mathbb{R} \setminus (-|\hat{\lambda}_{K+1}| - \Delta, |\hat{\lambda}_{K+1}| + \Delta)$. Now we view $Y$ as a perturbation of $\hat{Y}$, hence

$$R = \hat{L}Y - Y\hat{\Lambda} = \hat{L}Y - LY + (LY - Y\hat{\Lambda}) = (\hat{L} - L)Y \tag{11}$$

$$||R|| = ||(\hat{L} - L)Y|| \leq ||\hat{L} - L||||Y|| \leq \varepsilon. \tag{12}$$

From Theorem 13 the result follows. $\qquad\square$

**Proof of Proposition 6** For 1:

$$||F||_F^2 = \operatorname{trace} FF^T = \operatorname{trace} U\Sigma V^T V\Sigma U^T = \operatorname{trace} U^T U\Sigma V^T V\Sigma = \operatorname{trace} \Sigma^2$$

$$= 1 + \sum_{k=2}^{K} \cos^2\theta_k = 1 + \sum_{k=2}^{K}(1 - \sin^2\theta_k) = K - \sum_{k=2}^{K}\sin^2\theta_k \text{ since } \theta_1 = 0 \tag{13}$$

$$\geq K - (K-1)\varepsilon'^2 \tag{14}$$

For 2: Denote $\operatorname{trace}\hat{M}^T M = <\hat{M}, M>_F$. Then $||M - \hat{M}||_F^2 = ||M||_F^2 + ||\hat{M}||_F^2 - 2 < \hat{M}, M >_F \leq K + K - 2(K - (K-1)\varepsilon'^2) = 2(K-1)\varepsilon'^2$. $\qquad\square$

**Proof of Proposition 7** We have that $| < M - \hat{M}, M' - \hat{M} >_F | \leq ||M - \hat{M}||_F ||M' - \hat{M}||_F$. From Proposition 6 the r.h.s is no larger than $2(K-1)\varepsilon'^2$.

$$- < M - \hat{M}, M' - \hat{M} >_F \leq ||M - \hat{M}||_F ||M' - \hat{M}||_F \leq 2(K-1)\varepsilon'^2 \tag{15}$$

$$- < M, M' >_F + < \hat{M}, M >_F + < \hat{M}, M' >_F -||\hat{M}||_F^2 \leq 2(K-1)\varepsilon'^2 \tag{16}$$

$$< M, M' >_F \geq < \hat{M}, M >_F + < \hat{M}, M' >_F -K - 2(K-1)\varepsilon'^2 \tag{17}$$

$$\geq 2K - 2(K-1)\varepsilon'^2 - K - 2(K-1)\varepsilon'^2 = K - 4(K-1)\varepsilon'^2 \tag{18}$$

Now, note that $\operatorname{trace} MM' = \operatorname{trace} YY^T Y'(Y')^T = \operatorname{trace}((Y')^T Y)(Y^T Y') = ||Y^T Y'||_F^2$. Moreover, by (7), $Y_Z$ and $Y$ differ by a unitary transformation. Since $||\ ||_F$ is unitarily invariant, the result follows.

**Proof of Theorem 4** We apply Theorem 9 of [13] with $A_X = Z, A_{X'} = Z'$, and $\tilde{A}_X = Y, \tilde{A}_{X'} = Y'$. It follows that $p_{XY_{kk'}} = \sum_{i \in k \cap k'} \hat{d}_i / \sum_{i=1}^{n} \hat{d}_i$. Hence, the point weights are proportional to $\hat{d}_{1:n}$. Also, evidently, $p_{min}/p_{max} = \delta_0$, and the result follows.

Note that we use the fact that both PFM's have degrees equal to $\hat{d}_{1:n}$ to obtain this proof. $\qquad\square$

**Proposition 14** *Assumptions 3 and 4, imply $||\operatorname{diag}(\sin\theta_{1:K}(\hat{Y}, Y))|| \leq \varepsilon/|\hat{\lambda}_K^A| = \varepsilon'$, where $\hat{\lambda}_K^A$ is the $K$-th eigenvalue of $\hat{A}$.*

**Proof of Proposition 14** We consider $\hat{A}$ a perturbation of $A$, its eigenvectors $\hat{Y}$ as the perturbed eigenvectors of $A$ and $M = \hat{\Lambda}$. Then, $R = A\hat{Y} - \hat{Y}\hat{\Lambda}$

$$\|R\| = \|A\hat{Y} - \hat{Y}\hat{\Lambda}\| \tag{19}$$

$$= \|(A\hat{Y} - \hat{A}\hat{Y}) + (\hat{A}\hat{Y} - \hat{Y}\hat{\Lambda})\| \tag{20}$$

$$\leq \|(A - \hat{A})\hat{Y}\| \tag{21}$$

$$\leq \|A - \hat{A}\|\|\hat{Y}\| \leq \varepsilon. \tag{22}$$

The separation between $\hat{\Lambda}$ and the residual spectrum of $A$ is $|\hat{\lambda}_K|$. From the main Davis-Kahan theorem 13 the result follows. $\qquad\square$

**Proof of Proposition 8** The proofs of 1 and 2 are straightforward. To show 3, note that $A = ZC^{-1}Z^T\hat{A}ZC^{-1}Z^T = Y_Z C^{1/2}BC^{1/2}Y_Z^T = Y_Z U\Lambda U^T Y_Z^T = Y\Lambda Y^T$. The definition of $B$ above shows that this is the Maximum Likelihood estimator of $B$ given the clustering $\mathcal{C}$.

$$\Leftrightarrow \quad B_{kl} = \frac{\#\text{edges from cluster } k \text{ to cluster } l}{n_k n_l} \tag{23}$$

**Proof of Theorem 9** We now follow the steps outlined in section 3 with $\varepsilon'$ from Proposition 14 to obtain our main stability result.

**Proof of Proposition 10** In the Proof of Proposition 7, we replace the bounds corresponding to $< \hat{M}, M >_F, \|\hat{M} - M\|_F$ by the actual values computed from $M, \hat{M}$. We obtain

$$< M, M' >_F \geq < \hat{M}, M >_F - (K-1)(\varepsilon')^2 - 2\sqrt{2(K-1)}\varepsilon'\|\hat{M} - M\|_F. \tag{24}$$

**Proof of Proposition 3**

From the Proof of this theorem, we have that $\|L^* - \hat{L}\| = o(1)$, $\|(D^*)^{1/2} - \hat{D}^{1/2}\| = o(1)$, $\|\lambda^* - \hat{\Lambda}\| = o(1)$, and $\|\hat{Y} - Y^*\| = o(1)$. Let $Z$ be the indicator matrix of $\mathcal{C}^*$. The principal eigenvectors of $L^*$ are $Y^* = (D^*)^{1/2}Z(C^*)^{-1/2}$. It follows then that $\|Z^T\hat{D}Z - Z^TD^*Z\| = o(1)$, and since $C = Z^T\hat{D}Z, Y_Z = \hat{D}^{1/2}ZC^{-1/2}$ we have that $\|Y_Z - Y^*\| = o(1)$, $\|F^* - F\| = o(1)$ where $F^* = Y^TY^*$. Moreover, since $\|\hat{Y} - Y^*\| = o(1)$, $\|F - I\| = o(1)$ Hence $\|UV^T - I\| = o(1)$. Since the choice of $B$ depends only on $\mathcal{R}(Y_Z)$, it follows immediately that $\|BB^T\hat{L}B^TB - B^*(B^*)^TL^*(B^*)^TB^*\| = o(1)$. Now, $L = Y_Z UV^T\hat{\Lambda}VU^TY_Z^T + BB^T\hat{L}B^TB$, and $L^* = Y^*\Lambda^*(Y^*)^T + B^*(B^*)^TL^*(B^*)^TB^*$, which completes the proof. $\qquad\square$

**perturbation of the PFM model** To obtain a noisy PFM model $A$, we calculate the first $K$ piecewise constant [14] eigenvectors $V$ of the transition matrix $P = D^{-1}A$, from which we obtain $V^*$ by perturbing each entry in $V$ with a noise $\epsilon \sim unif(0, 10^{-4})$. The perturbed similarity matrix $A$ is then obtained as $A = D^{1/2}(D^{1/2}V^*\hat{\Lambda}V^{*T}D^{1/2} + \hat{Y}_{low}\hat{\Lambda}_{low}\hat{Y}_{low}^T)D^{1/2}$. An adjacency matrix $\hat{A}$ is generated from $A$. In figure 2, we show the perturbed graphs $A$ and $\hat{A}$.

$$A \qquad\qquad\qquad\qquad \hat{A}$$

Figure 2: Left: the visualization of the perturbed $A$. Right: the visualization of the perturbed $\hat{A}$