[Reviews · NeurIPS 2016]

Reviewer 1

Summary

The authors propose new theoretical results on graph clustering that do not make assumptions on the generation of the graph under study (the observed graph is not assumed to have been generated according to a specific generative model such as the stochastic block model). The main idea of the paper is that if a graph clustering method is able to discover a clustering that fits the graph reasonably (for a certain definition) then any other clustering that fits as well the graph should be close to the first one.

Qualitative Assessment

This paper is very interesting and addresses in my opinion a very important problem by providing distribution free results for graph clustering. While some of the content has been moved to the supplementary materials due to the page number restriction of the conference, the paper is quite clear and can be read without referring to this additional content. I'm not convinced that the use of "hats" on the actual observation (e.g. \hat{L} for the observed Laplacian) is a good choice, but apart from this detail, the paper is reasonably clear. The results themselves are very interesting and well commented and illustrated. They are however not without limitations. As shown in the experimental part, on some real world data, the graph has to be smoothed in order for some structure to be recoverable (in the sense of the theoretical framework developed by the authors), whereas many classical algorithms are able to recover the ground truth structure. In addition, and maybe more importantly, two limitations appear: 1) the main result for PFM uses a non standard distance measure between clusterings. The authors argue that this measure might be seen as more natural than the classical one, but this is questionable. 2) the results are based on a form of "proxy" model, in the sense that from a clustering, a model (e.g. a PFM) is reconstructed and this model is compared to the original graph (rather than whatever model that was used to obtain the clustering). This is very natural in the context chosen by the authors, but this has also the effect of constraining strongly the notion of what is a good clustering for a given graph (here, this means that the graph is well approximated in the normalized Laplacian sense). This is obviously needed to obtain actual results, but one might wonder what are the impact of those restrictions especially considering the limitations faced by the framework on some real world examples.

Confidence in this Review

2-Confident (read it all; understood it all reasonably well)


Reviewer 2

Summary

The paper gives a relationship between different distances of clustering. The paper considers the network models such as Preference Frame Model (PFM) and Stochastic Block Model (SBM) and proves a locally convex-like property for clustering. It is shown that for any two clusterings, which estimate the parameters of the model fitted to the network data equally well, are also near each other in terms of mis-classification distance measures.

Qualitative Assessment

The paper gives a nice property of clusterings in network data. The property of equivalence between the different measures of distance of clusterings for network data has been stated and proved in the paper. The paper gives both theoretical and experimental validations of the results stated in the paper. The results given in the paper are novel, but there has been recently some works of similar flavor in network literature like Peng, Sun and Zanetti (COLT, 2015), Chao, Lu, Ma, Zhou (Arxiv, 2015). The theoretical parts of the paper are also quite simple extensions of well-known results. As mentioned in the paper, the work can have some applications in detecting clusterability, but the formal results in that regard were not provided in the paper. The paper is well-written and explains the core ideas of the paper quite lucidly.

Confidence in this Review

2-Confident (read it all; understood it all reasonably well)


Reviewer 3

Summary

The manuscript presents a set of algorithms for clustering over graphs, and for testing whether two cluster assignments are similar.

Qualitative Assessment

As mentioned above, although the algorithms themselves are clear, their motivation and the reason why they appear as a list of algorithms is not well explained. The algorithms themselves are modifications of existing ones, and the only novelty is the constraints presented at the end.

Confidence in this Review

1-Less confident (might not have understood significant parts)


Reviewer 4

Summary

The paper presents a generic template for theoretical guarantee for the stability of a clustering C that fits the data well. The stability is measured in terms of the distance between C and any other C' that fit the same data well. The paper also provides specific example of using the template for two block models, PFM and SBM, and all the quantities involved in the guarantees can be computed explicitly from the data. Furthermore, it also uses the stability results thus obtained to show exact recovery guarantees for the models. However, the simulations suggest that either the goodness fit used or the conditions of the theorems provided render the results not too useful.

Qualitative Assessment

1) The paper considers the problem of providing generic template for theoretical guarantees for clustering without assuming that the data is from a specific model. The goal is to obtain such guarantees with quantities that can be computed from the data and the output of the clustering algorithms being compared. Providing such model free theoretical guarantees for clustering is of importance for both theoretical and practical purposes. 2) Given that PFM includes DC-SBM as a special class which in turn includes SBM, it is not clear why the two cases (PFM and SBM) are treated differently. Given that Spectral Clutering works well for all the models specified, why not use the same model estimator? In particular, it is not clear why the Laplacian is used for PFM while the adjacency matrix is used for the SBM. Also, the results for PFM is for weighted ME whereas for SBM it is in terms of ME. How do they compare with each other? Further, why are the results for DC-SBM not given? 3) In Proposition 10, what is the definition of < M', M >_F ? 4) In Assumption 5: Is it ||L^{hat}^2 - L^2||_F or ||L^{hat} - L||_F^2 ? If former, what is the intuition behind squaring the Laplacians? 5) The paper has focused mainly on disjoint clusters. Do the techniques/recipe used in this paper extend beyond block models to other models for which Spectral Clustering works? For example, Spectral clustering is known to work well for hierarchical clustering and there are theoretical guarantees available ( "Noise Thresholds for Spectral Clustering", S. Balakrishan et. al. 2011). How would the template work for this case? 6) The evaluation metric considered as well as the estimators considered both in the cases of PFM and SBM are limited to C and C' having the same number of clusters. Different clustering algorithms can potentially give different number of clusters. For example, let the eigenvalues of A might suggest K clusters for C and that of the Laplacian of A might suggest K' = K + 1 clusters for C' such that all the clusters in C and C' are the same except for one cluster that splits into two. C' is still "close" to C, but it cannot be captured by the given framework. 7) Related work section does not reflect a lot of model free results that exist: for example: "Clustering with Spectral Norm and the k-means Algorithm", Kumar and Kannan, 2010. "Clustering under Perturbation Resilience", Balcan et. al. 2012. "Clustering under Approximation Stability", Balcan et. al. 2013. How does the notion of stability considered in this work related to those in these works? In particular, how is the stability in this paper related to the stability measured for perturbation of centers for two clusterings? 8) The perturbations tolerated by simulations for PFM are too small. Line 252-253 casually states "large eigengap makes PFM(G, C) differ from PFM(G, C_0) even for very small perturbations" without really qualifying the statement. It is unclear why it is so. On the other had, the political blog data is smoothed out by raising the power of A which would further exaggerate the gaps in eigenvalues, but in this case it seems to help stability? Also, if such small perturbations can affect the model fit, then it renders the guarantee not of much use for practical purposes (which is one of the main selling points since the quantities can be actually calculated from the data). Lines 248-250 state that SBM guarantee conditions seem to be very hard to satisfy which is unfortunate. 9) Figure 1 caption mentions "For SBM, \delta is always greater than \delta_0". Which of the subplots is this referring to? If SBM is not plotted, then it should be made clear in the caption. 10) The guarantees in the theorems for PFM are for weighted ME, whereas in the experiment the perturbation is done for ME. Minor Points: 1) Labels for axes in Figure 1 are missing. 2) (Line 44) For block models, A_{ij} for i > j are independent _given the cluster assignments_ for i and j. 3) B, C and Y have been used with _different_ definitions for PFM and later in 3.2 for SBM. This can be confusing in general. Given that the paper has a lot of notations, it is better to maintain consistency throughout. For example, may be use Y_{\orthogonal} and C_D respectively instead of B and C for PFM. Also, for the estimator of SBM, Y_Z is not required. It is better to use the expression of Y_Z directly in Proposition 8. Similarly for \lambda in PFM and SBM, using \lamda^L an \lambda^B would clearly distinguish which eigenvalues are being talked about. 4) 4.4 is better suited to be a subsection in related work.

Confidence in this Review

2-Confident (read it all; understood it all reasonably well)


Reviewer 5

Summary

This paper provides very general concentration guarantees based on spectral clustering techniques for all types of community detection problems. They prove these guarantees for different generative models without having to use any of the parameters from those models. The work was interesting, relevant and advances a popular area of study in an important direction.

Qualitative Assessment

This paper posed a very interesting question about the stability of solutions for the problem of community detection. They phrased and proved these results in great generality while also focusing and applying them to specific popular graphical models of interest. This paper was also very well written and a easy and enjoyable read. They explained the motivation behind almost all of the ways they chose to define error and make assumptions required for their proofs. The only real criticism I had while reading this paper, which the authors themselves even point out in their conclusion, is that these proofs did depend on spectral algorithms and results which are not tight and are even almost intractable for many of the most interesting community detection problems when graphs are very sparse. That being said, this was a great start into a field of stability results for community detection that I hope and expect to be progressed even further in the near future.

Confidence in this Review

3-Expert (read the paper in detail, know the area, quite certain of my opinion)


Reviewer 6

Summary

In the paper the authors try to come up with a theory which works for graph clustering in model free framework. However, the result is naive and unconvincing.

Qualitative Assessment

In the paper the authors try to come up with a theory which works for graph clustering in model free framework. However, the result is naive and unconvincing. The authors pretend to construct a new theory but theit theory is just a naive handwaving. Just to give a couple of examples: Theorem 1 (Generic Theorem) ... "which also fits G well is close to C" ... Proposition 3 (Informal) ... "under standard recovery conditions" ...

Confidence in this Review

3-Expert (read the paper in detail, know the area, quite certain of my opinion)